# Modeling and Analysis of Maximum Power Tracking of a 600 kW Hydraulic Energy Storage Wind Turbine Test Rig

**Liejiang Wei [1,2], Peng Zhan [1], Zengguang Liu [1,2,*], Yanhua Tao [1] and Daling Yue [1,2]**

[1] Energy and Power Engineering College, Lanzhou University of Technology, Lanzhou 730050, China; weiliejiang@lut.edu.cn (L.W.); zhanpeng9234@163.com (P.Z.); 18335163909@163.com (Y.T.); yuedl@lut.edu.cn (D.Y.)

[2] Key Laboratory of Fluid Machinery and Systems (Gansu Province), Lanzhou 730050, China

* Correspondence: liuzg@lut.edu.cn; Tel.: +86-138-9336-4273

**Abstract:** An innovative wind turbine with a particular hydraulic transmission and energy storage system is proposed in this paper. The purpose of applying the hydraulic transmission is to remove the gearbox and power converter of traditional wind turbine and cooperate on wind resource storing with the energy storage system. To overcome the volatility and intermittence shortcomings of wind and improve the output power quality, hydraulic accumulators are used as the energy storage device for wind energy regulation. The original gearbox and generator in the nacelle of a Micon 600 wind turbine were removed and replaced with a hydraulic pump to make a test rig for the investigation into maximum power point tracking (MPPT) of this hydraulic wind turbine concept. The mathematical model of the entire test system is established according to the four function modules. The MPPT control strategy based on the tip speed ratio (TSR) is adopted and a control system containing three closed-loop controls is designed to achieve maximum wind power extracting and produce constant frequency power generation. Ultimately, the dynamic response of rotor speed control is revealed under step change of wind speed and the maximum power tracking performance of the 600 kW hydraulic energy storage wind turbine test bench is simulated and analysed by subjecting to turbulent speed condition. The simulation results demonstrate that the rotor of the wind turbine can run at the expected optimal speed depending on wind speed, and the wind power utilization coefficient of the unit is stabilized at about the maximum value.

**Keywords:** hydraulic wind turbine; maximum power point tracking; energy storage system; tip speed ratio

## 1. Introduction

Wind energy is the most reliable and developed renewable energy source over past decades. 2018 was still a good year for the global wind industry with 51.3 GW of new wind energy installed, which is also the fourth year that annual installations have topped 50 GW each year since 2014 [1]. Hydraulic wind turbine will be a great potential for reducing the cost and improving power qualities of wind power generation. This is because the hydraulic wind turbine, which eliminates the high fault gearbox and expensive power converter, create significant advantages to wind turbine, reducing the nacelle weight, lowering installation cost, simplifying maintenance and also facilitating wind energy storage [2–4].

The researchers throughout the world have studied the hydraulic wind turbine from different perspectives and built various experiment platforms to prove the stability and reliability of their research results. Based on the 1 MW semi-physical simulation experimental platform, as shown in

Figure 1a, Aachen University of Technology in Germany compared the system behavior of a hydrostatic transmission for wind turbines with that used in mobile applications [5,6]. Afshin Izadian et al. derived the mathematical modeling of the hydraulic wind power transfer system and demonstrated the dynamic behavior with the hydraulic wind turbine experimental setup that was described in Figure 1b [7,8]. C Ai from Yanshan University established the energy transfer and dissipation models of hydraulic wind turbine and analyzed the change law of energy transfer. The 30 kVA hydraulic wind turbine simulation platform (Figure 1c) was used to verify the accuracy of theoretical analyses [9]. The TRL4 System, which was very similar to the hydraulic wind turbine, was designed and constructed by NASA-JPL to simulate a hydraulic tidal energy turbine. The efficiencies of the pump, motor, and hydraulic power transfer of this unit were put forward and proved. The schematic of the TRL4 system was displayed by Figure 1d [10]. Biswaranjan Mohanty presents a dynamical model to investigate the performance of a hydraulic wind turbine coupled to the electrical grid through a synchronous generator. The NREL AOC 15/50 turbine equipped with a hydrostatic transmission is selected as research subject [11]. These experimental studies were all carried on in indoor laboratory and wind energy was realized by various simulators.

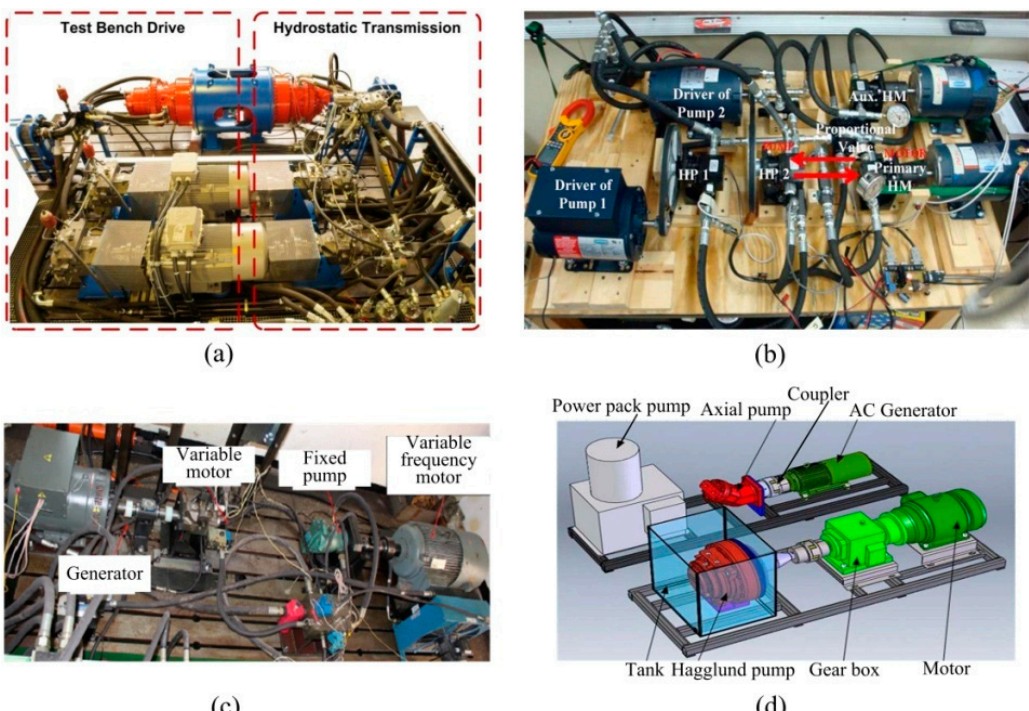

**Figure 1.** The experiment platforms for hydraulic wind turbine. (**a**) The simulation experimental platform in Aachen University of Technology; (**b**) the hydraulic wind turbine experimental setup in the Purdue School of Engineering and Technology; (**c**) the 30 kVA wind power test bench in Yanshan University; (**d**) the hydraulic tidal energy turbine test unit from NASA-JPL.

Utilizing the maximum power from the wind and outputting electrical energy as much as possible are the major problems of wind power generation [12]. The output power of variable speed wind turbine can be divided into four different working regions, as demonstrated in Figure 2. The first region, which is below the cut-in wind speed, is at the far left of the graph. The region at the far right is defined as the fourth, in which the wind speed is above the cut-out wind speed. In these two regions, the turbine must not work for safety and should be disconnected from the power grid [13]. In the third region between the rated wind speed and cut-out wind speed, the blades pitch angle is used as a control variable to restrict the mechanical power of the rotor to the rated power and avoid damaging to the turbine [14]. The remaining region between the cut-in wind speed and rated wind speed is the second, in which the power produced by wind turbine rotor will change with wind speed. In order

to achieve the maximal energy extraction from wind energy in the second region, the pitch angle of blades is kept to the minimum and the wind turbine has to operate at the maximum power absorption by maximum power point tracking (MPPT) methods [15]. In the literature, the commonly used MPPT techniques comprise tip speed ratio (TSR) [16], power signal feedback (PSF) [17], hill climb search (HCS) [18]. These three MPPT algorithms have different control variables and their advantages and drawbacks, but the dynamic performance of rotor speed control has an important influence on the application of them [19].

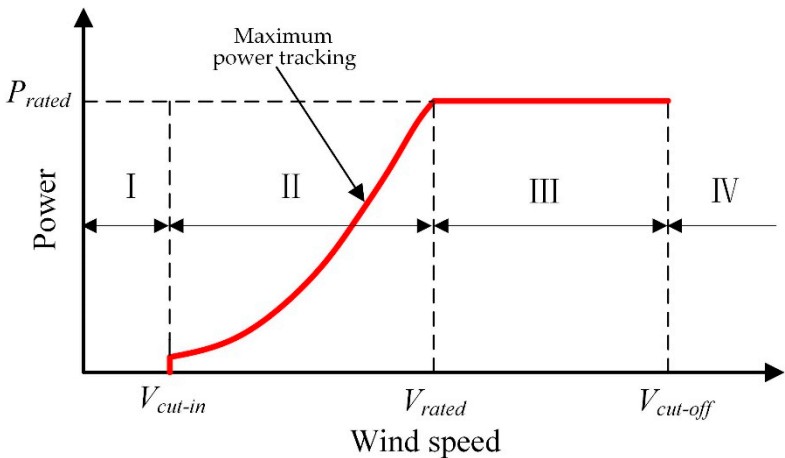

**Figure 2.** The working region of wind turbine.

Our research group has put forward an innovative hydraulic energy storage wind turbine idea in which a distinctive hydraulic transmission and large-capacity hydraulic accumulator are adopted to realize the light mass, simple structure, lower maintenance of the nacelle in this wind turbine and diminish or eliminate the influence from the fluctuations of wind energy on the quality of electrical power generated by wind farm. A 600 kW hydraulic energy storage wind turbine test rig using the proposed idea is being built and will be used in outdoor and real wind conditions to confirm the feasibility of the new idea and prove the correctness of modeling and simulation research on this hydraulic wind turbine.

This paper establishes the numerical model of the testing system and employs the TSR method to analyze and predict the maximum wind energy capture performance of the test rig and the hydraulic energy storage wind turbine proposal. Section 2 details the system configurations and function modules of the hydraulic energy storage wind turbine. In Section 3, the mathematical models of all components are presented. In Section 4, the operating principle of MPPT control technology based on the TSR is clearly stated, and the maximum power tracking and the constant frequency electric power generation of the hydraulic wind turbine are achieved through three closed-loop controls. The results and discussion of the simulation are performed in Section 5. Finally, the conclusions are drawn in Section 6.

## 2. System Overview

The scheme of hydraulic energy storage wind turbine investigated in this paper is shown in Figure 3. A main hydraulic pump is placed in the nacelle and directly coupled to the rotor of the wind turbine. A low speed high torque hydraulic pump that has only fixed displacement has to be selected for the low working speed of wind turbine. The main pump driven by the wind turbine rotor sucks the hydraulic oil in the low-pressure lines and expels the pressurized oil into a main hydraulic motor. The main hydraulic motor is coaxially connected to the energy storage hydraulic pump, and the former provides the rotational speed for the latter during normal operation. Finally, the wind energy captured by the rotor is converted into hydraulic energy by the energy storage hydraulic pump.

The high-pressure oil from energy storage hydraulic pump can enter the accumulator for energy storage, or directly drive the power generation hydraulic motor and synchronous generator to generate electricity, or both. In comparison with the conventional wind turbine, there is only one hydraulic pump in the nacelle and the other components are located on the ground, which make it possible to reduce the top mass. Another significant advantage of the hydraulic energy storage wind turbine is that the hydraulic accumulator can solve the coupling problem between the wind power captured by wind turbine rotor and electrical power produced by synchronous generator and compensate the imbalance of them automatically.

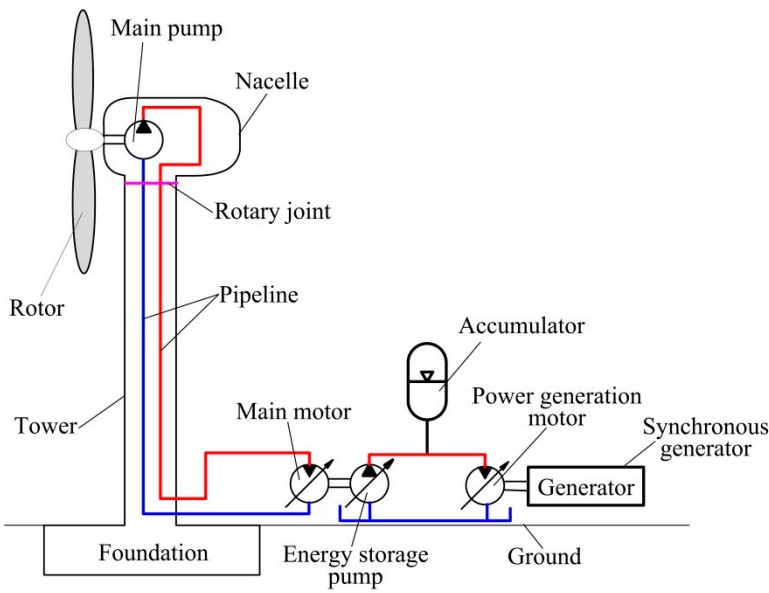

**Figure 3.** The overall scheme of hydraulic energy storage wind turbine.

The hydraulic energy storage wind turbine can be divided into four parts according to their own function, as shown in Figure 4. They are: (1) Wind turbine, (2) hydraulic variable transmission, (3) hydraulic energy storage, (4) electric power generation. Wind energy will be transformed into mechanical energy by the rotor of wind turbine. Hydraulic variable transmission plays a role of converting low speed of wind turbine rotor to high speed of main motor to meet the working speed of the energy storage pump. Wind energy will be converted into hydraulic energy and stored into the hydraulic accumulator in the hydraulic energy storage part, and the last part completes the change from hydraulic energy to electric energy.

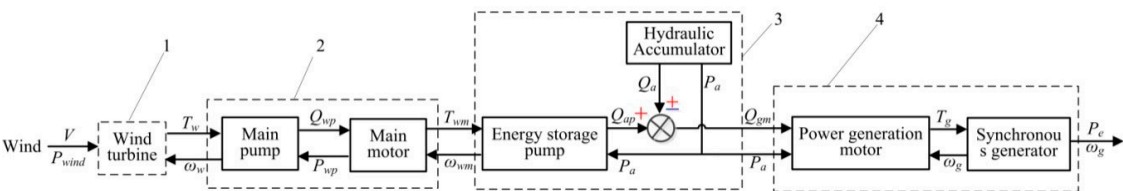

**Figure 4.** The function modules of hydraulic energy storage wind turbine.

## 3. Mathematical Model

Figure 4 also demonstrates the interactional relations and physical quantities between every component. According to the following government equation, the models of all components are created in the following sections.

### 3.1. Wind Turbine

Wind power is absorbed and converted into mechanical energy by the wind turbine rotor. The available wind power $P_{wind}$ and the actual mechanical power $P_w$ of the wind turbine rotor are expressed as follows:

$$P_{wind} = \frac{1}{2}\rho A v_w{}^3 \tag{1}$$

$$P_w = C_p P_{wind} \tag{2}$$

where $\rho$ is the air density, $A$ is the swept area of the rotor blades. $v_w$ is the wind speed. $P_w$ is related to $P_{wind}$ via a power coefficient $C_P$ of the wind turbine, which is a function of the tip speed ratio $\lambda$ and the pitch angle $\beta$. The typical curve representing the relationship between $C_P$, $\lambda$ and $\beta$ is shown in Figure 5. It is easy to see that the maximum $C_P$ is achieved when the tip speed ratio $\lambda$ is at the optimal value $\lambda_{opt}$ and the pitch angle $\beta$ is equal to zero [20].

$$C_p = 0.5176(\frac{116}{\lambda_i} - 0.4\beta - 5)e^{-\frac{21}{\lambda_i}} + 0.0068\lambda \tag{3}$$

$$\frac{1}{\lambda_i} = \frac{1}{\lambda + 0.08\beta} - \frac{0.035}{\beta^3 + 1} \tag{4}$$

$$\lambda = \frac{R\omega_w}{v_w} \tag{5}$$

the tip speed ratio $\lambda$ is the quotient of the peripheral velocity of the wind turbine rotor to the wind velocity, which is obtained from formula 5. $R$ is the radius of the wind turbine rotor. $\omega_w$ is the rotor angular speed of wind turbine rotor. The output torque of the rotor $T_W$ can be calculated as:

$$T_w = \frac{P_w}{\omega_w} \tag{6}$$

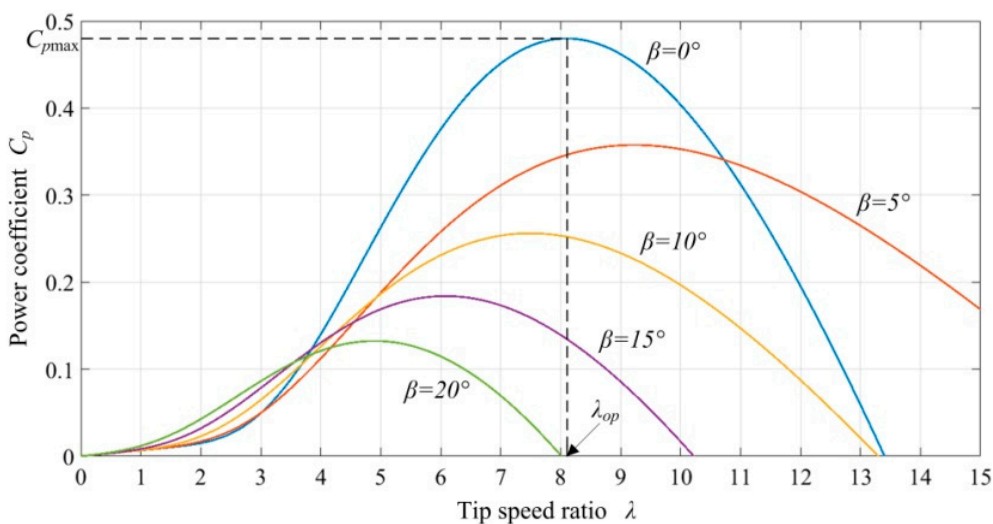

**Figure 5.** A typical relationship between the power coefficient and the tip-speed ratio.

## 3.2. Hydraulic Variable Transmission

### 3.2.1. Main Hydraulic Pump

Since the main hydraulic pump is coaxially connected to the wind turbine rotor, they have the same speed. The output flow of the main hydraulic pump $Q_{wp}$ and the driving torque of the hydraulic pump $T_{wp}$ are expressed as follows [21]:

$$Q_{wp} = V_{wp}\omega_{wp}\eta_{pv} \tag{7}$$

$$T_{wp} = V_{wp}(P_{wpo} - P_{wpi})/\eta_{pm} \tag{8}$$

where $V_{wp}$ is the theoretical displacement of the main hydraulic pump. $\omega_{wp}$ is the speed of the main hydraulic pump. $P_{wpi}$ and $P_{wpo}$ are the inlet pressure and outlet pressure of the main hydraulic pump, respectively. $\eta_{pv}$ and $\eta_{pm}$ represent the volumetric and mechanical efficiencies of the main hydraulic pump, which are generally considered to be constant. The dynamic equations of the wind turbine rotor and the main hydraulic pump are:

$$J_w\frac{d\omega_w}{dt} = T_w - T_{wf} - T_{wp} = T_w - B_w\omega_w - T_{wp} \tag{9}$$

where $J_w$ is the moment of inertia of the wind turbine rotor and main hydraulic pump. $T_w$ is the output torque of wind turbine. $T_{wf}$ and $B_w$ represent the damping torque and the damping coefficient of the wind turbine rotor and main hydraulic pump, respectively.

### 3.2.2. Main Hydraulic Motor

The main hydraulic motor and power generation hydraulic motor both are the variable displacement motor. The variable mechanism is a small and typical electro-hydraulic position servo system of a valve-controlled cylinder (shown in Figure 6). A first-order model can be used to describe the dynamic characteristics of the variable mechanism [22]:

$$\frac{D_m(s)}{U_m(s)} = \frac{K_m}{T_m s + 1} \tag{10}$$

where $D_m$ is the displacement of the variable motor. $U_m$ is the control signal to the servo controller. $K_m$ is the motor stroke-control gain and $T_m$ is the time constant of the variable mechanism.

$$Q_m = D_m\omega_m + C_{im}(P_1 - P_2) + C_{em}P_1 \tag{11}$$

where $Q_m$ is the motor flow and $\omega_m$ is the rotational speed of the motor. $C_{im}$ and $C_{em}$ represent the internal and the external leakage coefficients of the motor, respectively. The dynamic equation of the motor motion is established as follows using Newton's second law:

$$D_m(P_1 - P_2) = J_t\frac{d\omega_m}{dt} + B_v\omega_m + T_f + T_d \tag{12}$$

where $J_t$ is the total inertia of the entire rotor. $B_v$ is the coefficient of viscous friction of the rotor. $T_f$ is the coulomb friction torque. $T_d$ is the disturbance to the motor. $P_1$ and $P_2$ represent the inlet pressure and outlet pressure of the motor.

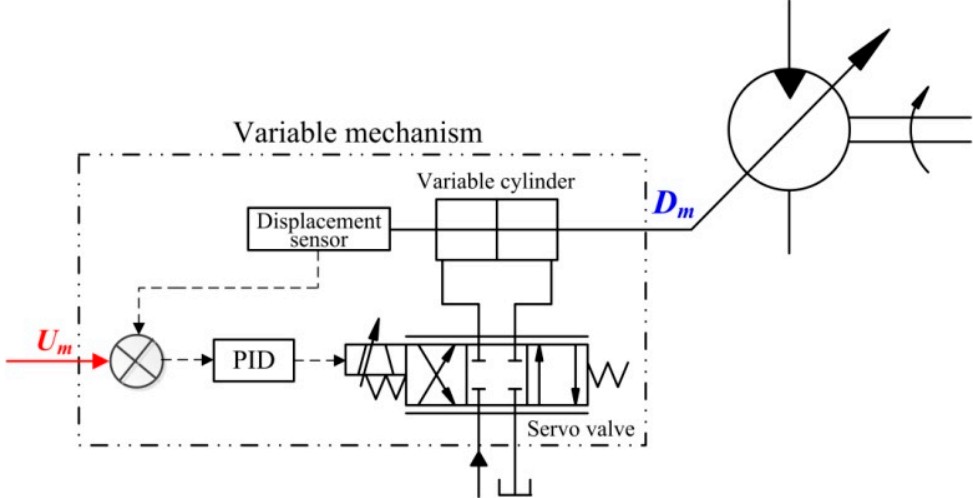

**Figure 6.** The schematic diagram of variable mechanism.

### 3.3. Hydraulic Energy Storage

#### 3.3.1. Energy Storage Hydraulic Pump

The energy storage hydraulic pump is variable displacement such as the main hydraulic motor. The structural composition and working principle of the variable mechanism are the same. Therefore, the analysis of the variable mechanism in energy storage hydraulic pump adopts the same modeling as the main hydraulic motor. The output flow of the energy storage hydraulic pump $Q_{ap}$ can be written as:

$$Q_{ap} = V_{ap}\omega_{wm}\eta_{apv} \tag{13}$$

The driving torque of the energy storage hydraulic pump $T_{ap}$ can be written as:

$$T_{ap} = V_{ap}(P_{apo} - P_{api})/\eta_{apm} \tag{14}$$

where $V_{ap}$ is the displacement of the energy storage hydraulic pump. $\omega_{wm}$ is the main hydraulic motor speed. $\eta_{apv}$, $\eta_{apm}$ are the volumetric efficiency and mechanical efficiency of the energy storage hydraulic pump. $P_{api}$ and $P_{apo}$ represent the inlet pressure and outlet pressure, respectively. The dynamic equations of the main hydraulic motor and the energy storage hydraulic pump can be written as:

$$J_{wm}\frac{d\omega_{wm}}{dt} = T_{wm} - T_{wmf} - T_{ap} = T_{wm} - B_{wm}\omega_{wm} - T_{ap} \tag{15}$$

where $J_{wm}$ is the inertia of the main hydraulic motor and the energy storage hydraulic pump. $\omega_{wm}$ and $T_{wm}$ are the speed and output torque of the main hydraulic motor. $T_{wmf}$ and $B_{wm}$ represent the damping torque and damping coefficient of the main hydraulic motor and the energy storage hydraulic pump; $T_{ap}$ is the driving torque of the energy storage hydraulic pump.

#### 3.3.2. Hydraulic Accumulator

The composition and operating principle of the hydraulic accumulator are displayed in Figure 7. The oil in the line flows into the shell of the accumulator when the accumulator is charged. Contrarily, the oil is squeezed into the line by the bladder. The compression of the gas in the bladder is assumed to obey the ideal gas law, the relationship between gas pressure $P_a$ and gas volume $V_a$ is expressed by the ideal gas Equation [23]:

$$P_a = \frac{P_0 V_0{}^n}{V_a{}^n} = \frac{P_0 V_0{}^n}{\left(V_0 + \int Q_a dt\right)^n} \tag{16}$$

$$Q_a = -\frac{V_0 P_0^{\frac{5}{7}}}{1.4 P_a^{\frac{12}{7}}} \frac{dP_a}{dt} \tag{17}$$

where $P_0$ and $V_0$ represent the pre-charge pressure and the initial volume of the gas, respectively. $Q_a$ represents the accumulator flow. $n$ is the polytropic index. In an isothermal compression, $n$ equals one. In an adiabatic compression, $n$ equals one point four that is used in this paper.

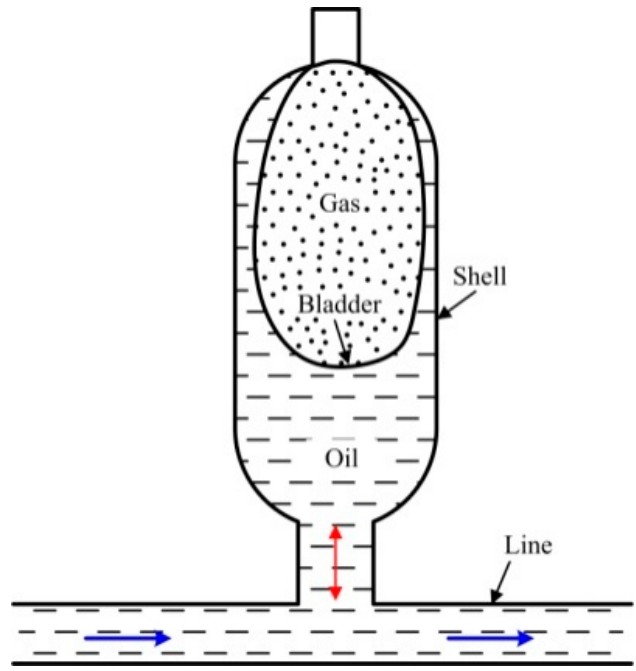

**Figure 7.** The schematic drawing of hydraulic accumulator.

### 3.4. Electric Power Generation

Synchronous Generator

A three-phase AC synchronous generator is used in this hydraulic wind turbine to the output electrical energy. The stator and rotor circuits diagram of synchronous generator are demonstrated in Figure 8. According to Park transformation and Ohm's law, the voltage balance equations of the rotor and stator in dq0 coordinates are computed as follows [24]:

$$u_{sd} = \frac{d\psi_{sd}}{dt} - \omega_e \psi_{sq} - R_a i_{sd} \tag{18}$$

$$u_{sq} = \frac{d\psi_{sq}}{dt} + \omega_e \psi_{sd} - R_a i_{sq} \tag{19}$$

$$0 = \frac{d\psi_{dd}}{dt} + R_{dd} i_{dd} \tag{20}$$

$$0 = \frac{d\psi_{dq}}{dt} + R_{dq} i_{dq} \tag{21}$$

where $u_{sd}$, $i_{sd}$ represent the stator voltage and current on the Park's $d$ axis. $u_{sq}$, $i_{sq}$ represent the stator voltage and current on the Park's $q$ axis. $R_a$, $R_{dd}$ and $R_{dq}$ represent the stator winding resistance and the damper winding's resistance. $\psi_{sd}$ and $\psi_{sq}$ are the stator windings flux linkage. $\psi_{dd}$ and $\psi_{dq}$ are the damper windings flux linkage. $i_{dd}$ and $i_{dq}$ are the stator current of the damper windings.

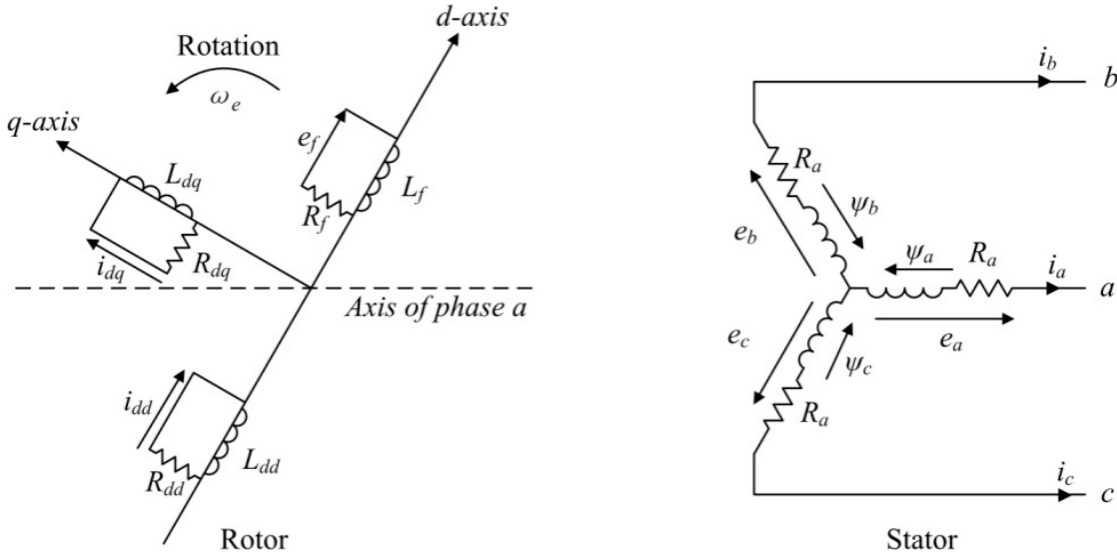

**Figure 8.** The stator and rotor circuits of synchronous generator.

The flux linkage equation of all winding is constituted as follows. The electromagnetic torque $\Gamma$ is computed by Equation (27):

$$\psi_{sd} = L_{sd}i_{sd} + \sqrt{\frac{3}{2}}M_{sf}i_{fl} + \sqrt{\frac{3}{2}}M_{sd}i_{dd} \tag{22}$$

$$\psi_{sq} = L_{sq}i_{sq} + \sqrt{\frac{3}{2}}M_{sq}i_{dq} \tag{23}$$

$$\psi_{fl} = L_{f}i_{fl} + \sqrt{\frac{3}{2}}M_{sf}i_{sd} + M_{fd}i_{dd} \tag{24}$$

$$\psi_{dd} = L_{dd}i_{dd} + \sqrt{\frac{3}{2}}M_{sd}i_{sd} + M_{fd}i_{fl} \tag{25}$$

$$\psi_{dq} = L_{dq}i_{dq} + \sqrt{\frac{3}{2}}M_{sq}i_{sq} \tag{26}$$

$$\Gamma = p(\psi_{sd}i_{sq} - \psi_{sq}i_{sd}) \tag{27}$$

where $M_{sf}$,$M_{sd}$,$M_{sq}$ and $M_{fd}$ represent the mutual inductance coefficient between the two windings. $L_{sd}$, $L_{sq}$, $L_{f}$, $L_{dd}$ and $L_{dq}$ are the inductance coefficient of all windings. $p$ represents the pole pairs. The dynamic equation of the generator hydraulic motor running synchronous generator is:

$$J_g\frac{d\omega_g}{dt} = T_{gm} - T_{gf} - T_e = T_{gm} - B_g\omega_g - T_e \tag{28}$$

where $J_g$ is the moment of inertia of the power generation motor and the synchronous generator. $\omega_g$ and $T_e$ represent the speed and the electromagnetic moment of the synchronous generator, respectively. $T_{gm}$ is the output torque of the power generation motor. $T_{gf}$ and $B_g$ are the damping torque and damping coefficients of the power generation motor and the synchronous generator.

## 4. System Control Scheme

### 4.1. Tip Speed Ratio (TSR) MPPT Algorithm

The generic TSR method of MPPT control for wind turbine is shown in Figure 9a. Using the optimal TSR $\lambda_{opt}$ and wind speed $v_w$, the TSR technique generates the optimum wind turbine rotor speed $\omega_{opt}$, at which wind power extracted by wind turbine rotor is maximized. Actual wind turbine rotor speed $\omega_w$ is measured to compare with the optimum wind turbine rotor speed. The deviation between them is transmitted to the controller and actuator system to alter the mechanical torque acting on wind turbine. Under the aerodynamic torque and mechanical torque, wind turbine rotor speed and power coefficient reach the optimum value simultaneously. The actuator system adjusts its output mechanical torque by reference signal, which has entirely different compositions and operating principle to various wind turbines. The output of the actuator system in the traditional wind turbine is the electromagnetic torque of the generator, which is used to balance with aerodynamic torque. Figure 9b shows the schematic diagram of the actuator system in this hydraulic energy storage wind turbine. As can be seen from the graph, the driving torque of the energy storage pump $T_{mm}$ is key to achieving the speed control of the wind turbine rotor, which is related to accumulator pressure $P_a$ and energy storage pump displacement $V_p$.

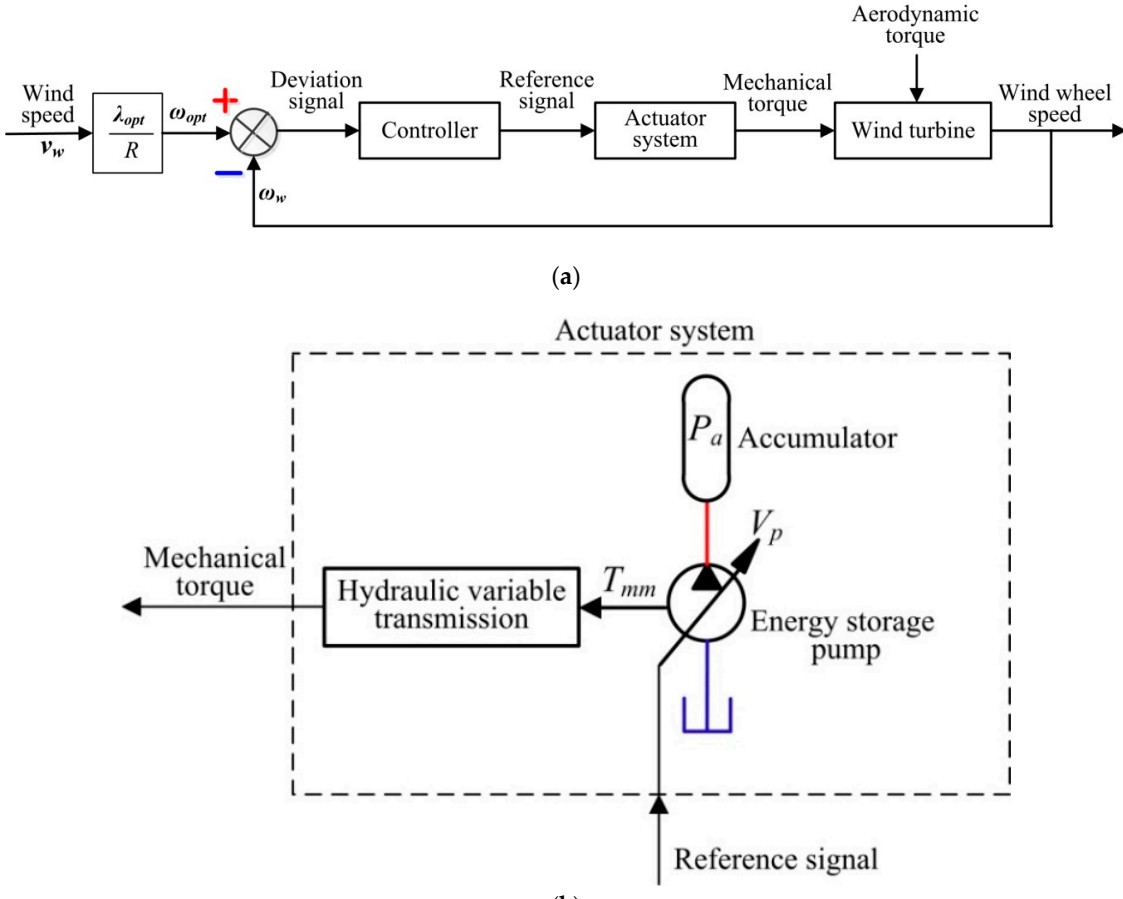

**Figure 9.** Maximum power point tracking (MPPT) algorithm based on tip speed ratio (TSR) of hydraulic energy storage wind turbine. (**a**) MPPT algorithm based on TSR of wind turbine; (**b**) actuator system of hydraulic energy storage wind turbine.

### 4.2. MPPT and Constant Frequency Control

The MPPT based on TSR and constant frequency control principle diagram of the hydraulic energy storage wind turbine is shown in Figure 10. The whole control system is made up of three closed loops: Maximum power tracking control, energy storage pump speed control, and generator speed control. The speed control of the wind turbine rotor is only achieved by adjusting the displacement of the energy storage pump because the accumulator pressure is determined by the charge and discharge processes of the hydraulic accumulator. According to Figure 9, the speed closed-loop of maximum power tracking control is established by connecting wind speed, wind turbine rotor speed, and the viable mechanism of the energy storage pump. The purpose of the energy storage pump speed control is to restrict the main motor speed and provide a normal working speed for the energy storage pump. In the last loop, the constant speed control of the synchronous generator is accomplished by regulating the displacement of the power generation motor, and the synchronous generator can work stably at the synchronous speed to generate constant frequency power.

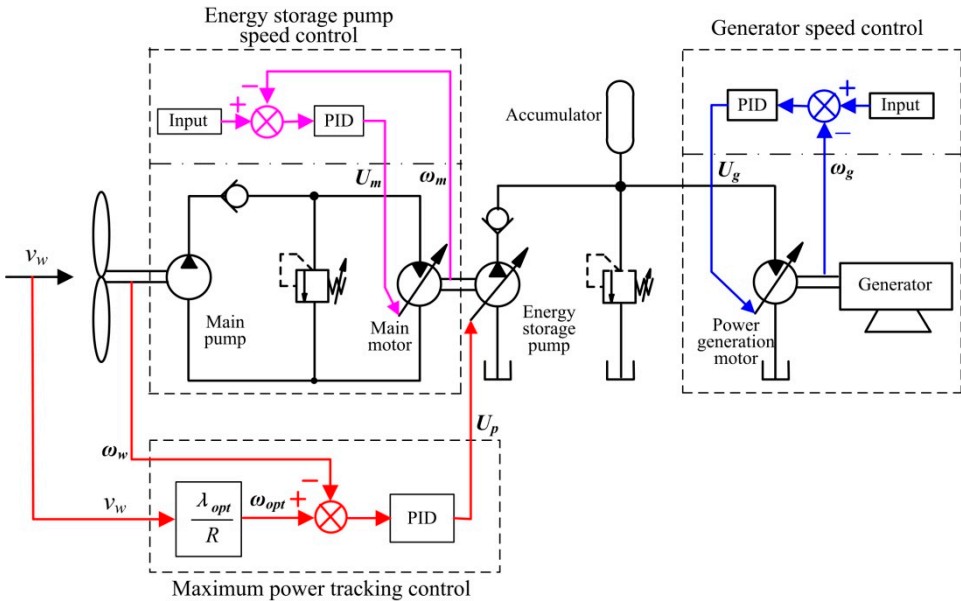

**Figure 10.** MPPT and constant frequency control of hydraulic energy storage wind turbine.

## 5. Simulation and Discussion

### 5.1. The Design of the Experimental Prototype

An obsolescent Micon 600 wind turbine (NEG Micon, Randers, Denmark) worked in the Huitengxile wind farm in Inner Mongolia is chosen and turned into the experimental platform of this hydraulic energy storage wind turbine concept. The whole layout plan of the experimental platform based on the 3D CAD design software SOLIDWORKS (Dassault Systemes, Massachusetts, USA) is shown in the middle of Figure 11. The rotor blades of the Micon 600 wind turbine located on the left side of the diagram below remain unchanged for capturing wind energy. The internal structure of the altered nacelle is displayed on top of Figure 11. After being processed, the main shaft is directly connected to the main hydraulic pump by means of a spline. The newly-designed brake system is also mounted on the main shaft to stop the wind turbine rotor from turning. Thirty hydraulic bladder accumulators form the accumulator group to conduct the wind energy storage, which have a total capacity of 6000 litres. To avoid a great energy loss for a long pipeline, the installation position of the motor and generator is close to the accumulator group. The load bank, which is shown on the lower right, is used to consume the electrical energy from the asynchronous generator.

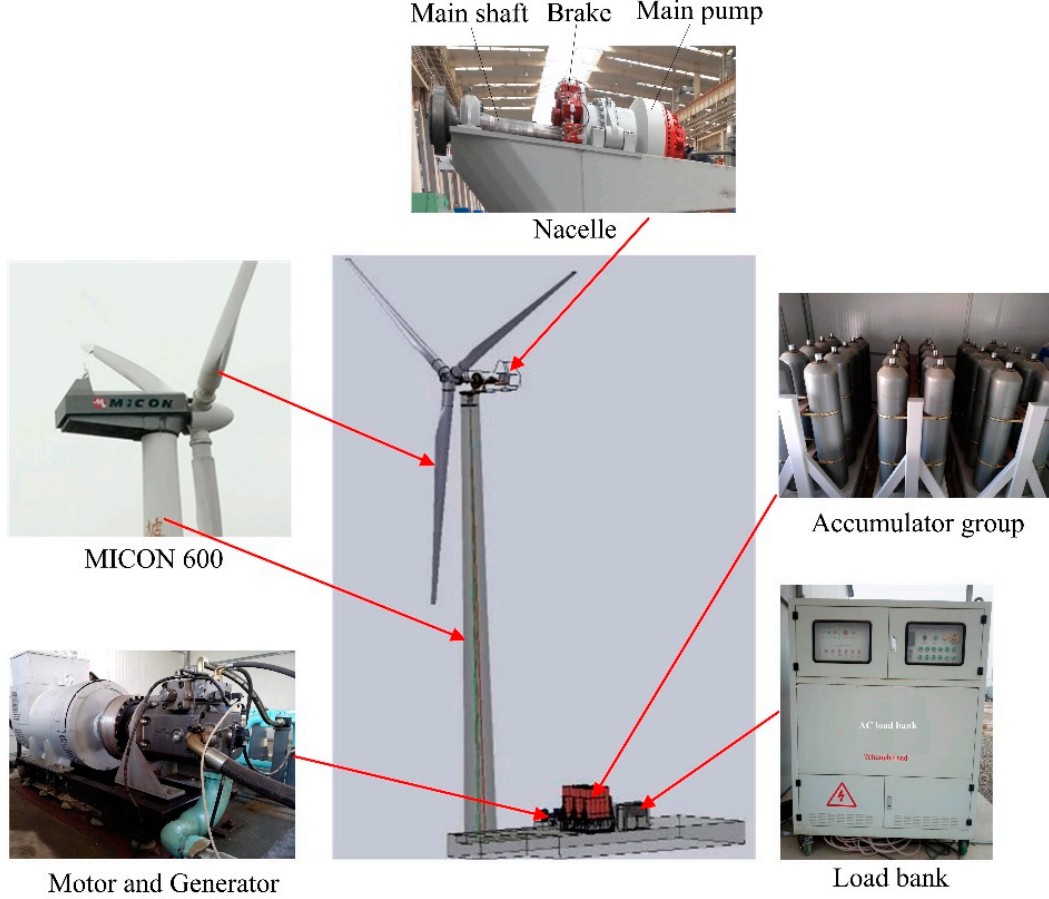

**Figure 11.** The layout design of 600 kW hydraulic energy storage wind turbine test rig.

According to the MPPT and constant frequency control principle, as shown in Figure 10, the whole simulation model of 600 kW hydraulic energy storage wind turbine test rig is established at the software platform of MATLAB (version 6.5, MathWorks Company, Natick, MA, USA). The main parameters are displayed in Table 1.

**Table 1.** The simulation parameters of the 600 kW hydraulic energy storage wind turbine test rig.

| Parameter | Value | Unit |
|:---:|:---:|:---:|
| Rotor diameter | 44 | m |
| Air density | 1.225 | kg/m$^3$ |
| Rotor equivalent viscosity efficient | 50 | N·m(r/min) |
| Rotor equivalent moment of inertia | 20,000 | kg·m$^2$ |
| Main hydraulic pump placement | 55,300 | mL/r |
| Main motor displacement | 1000 | mL/r |
| Main motor equivalent viscosity efficient | 0.1 | N·m(r/min) |
| Main motor equivalent moment of inertia | 40 | kg·m$^2$ |
| Accumulator capacity | 6000 | L |
| Accumulator initial oil pressure | 180 | bar |
| Accumulator gas pre-charge pressure | 100 | bar |
| Energy storage pump displacement | 500 | mL/r |
| Generation motor speed command | 1500 | r/min |

### 5.2. Simulations and Analyses

In order to analyze the effectiveness and dynamic characteristics of the wind turbine rotor speed control and evaluate the MPPT performance of the hydraulic energy storage wind turbine test rig in advance, the performance simulations of the model are executed under two different wind speed conditions, step and turbulent wind speed condition. The step input is ideal for inspecting the dynamics of the automatic control system. The turbulent wind condition is employed under MPPT control because the wind of wind farm is characterized by turbulence and instability.

#### 5.2.1. Step Wind Speed Condition

Figure 12 shows the wind speed profile, which has a step change in cycle. The step change wind speed is employed to obtain the dynamic response of wind turbine rotor speed control. The averages of the minimum and maximum wind speeds are 6 m/s and 8 m/s, respectively, and the turnaround is 50 s.

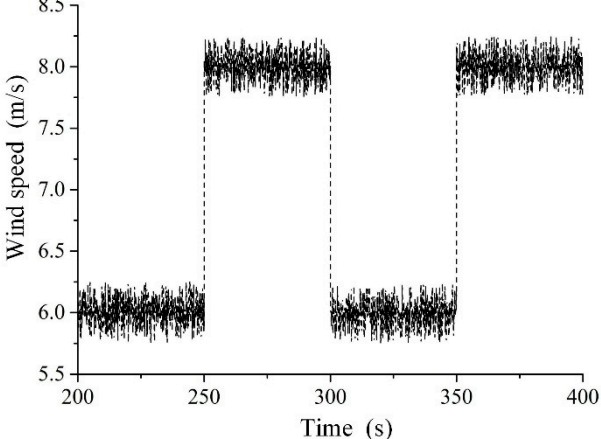

**Figure 12.** Step wind speed.

Figure 13a accurately reflects the comparison between the actual speed of the wind turbine rotor and the optimal speed. When the wind speed is 6 m/s and 8 m/s, the optimum rotor speed is 20 r/min and 28 r/min. The actual rotor speed closely follows the optimal speed, but due to the large rotary inertia of the wind turbine rotor, there is a slight overshoot and delay between them. The speed error between the two speeds can be seen in Figure 13b. The speed error reaches the maximum value when the wind speed step occurs, which is about 7 r/min.

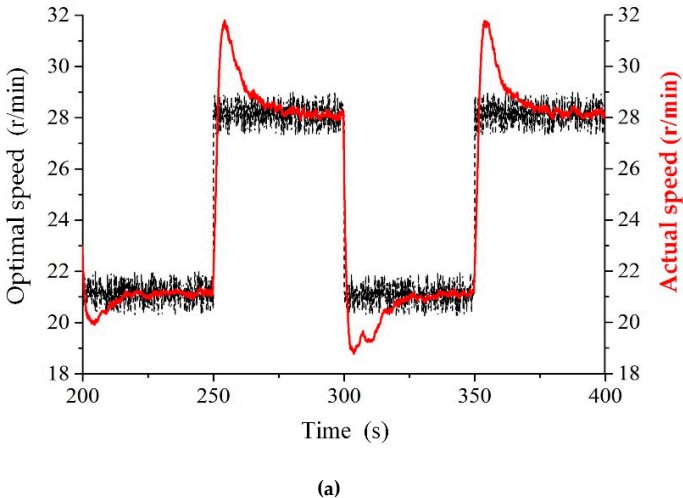

**(a)**

**Figure 13.** *Cont.*

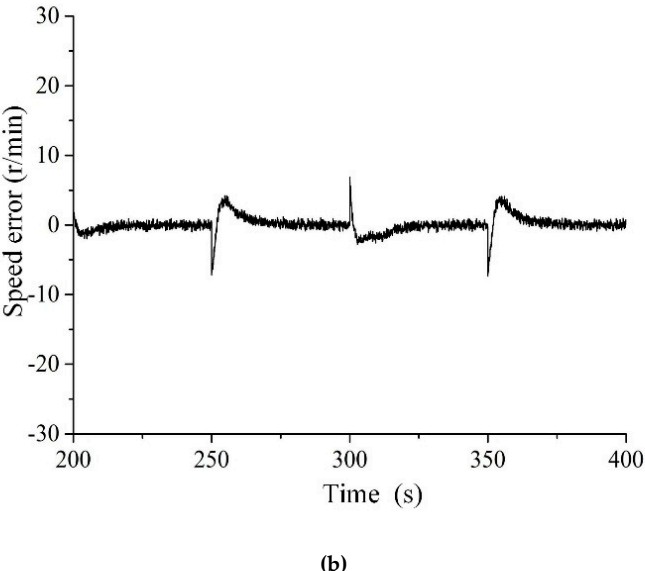

**(b)**

**Figure 13.** The optimal speed and actual speed of wind turbine rotor. (**a**) Speed comparison curve; (**b**) speed error between the optimal speed and actual speed.

Figure 14 shows the relationship between the energy storage pump displacement coefficient and the main motor. It can be seen from the graph that both displacement coefficients have the same trends of evolution. However, the displacement of the energy storage pump has a minor fluctuation. That is because the wind turbine is designed to operate at the optimum tip speed ratio by continuously adjusting the displacement of the energy storage pump. At the same time, it can be known that the displacement curve of the energy storage pump has a short overshoot with the wind speed step change. It is making the wind turbine rotor with a large inertia decelerate or accelerate suddenly. When the displacement coefficient of the main motor decreases from 0.37 to 0.1, the displacement coefficient of the energy storage pump reduces from 1 to 0.77.

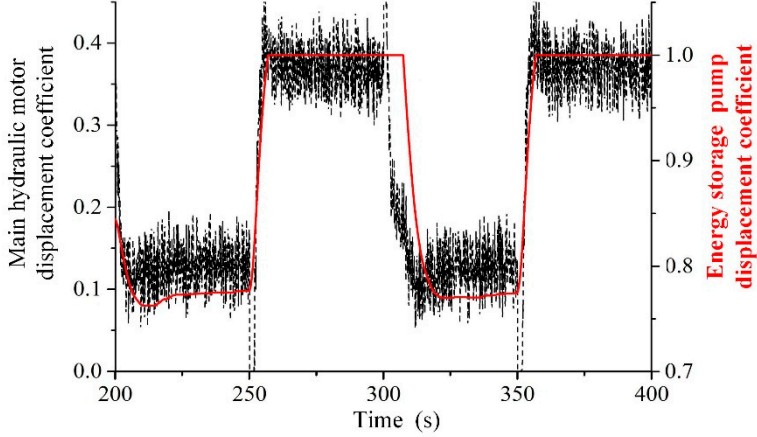

**Figure 14.** Displacement coefficient of energy storage pump and main motor.

Figure 15 depicts the comparison between the wind turbine rotor speed and the speed of the energy storage hydraulic pump. At step speeds of 6 m/s and 8 m/s, the wind turbine rotor speed is continuously switched between 21 r/min and 28 r/min. However, the speed of the energy storage pump can basically maintain its normal working speed of 1500 r/min, which is achieved by PID (Proportion Integral Differential) adjustment of the main motor.

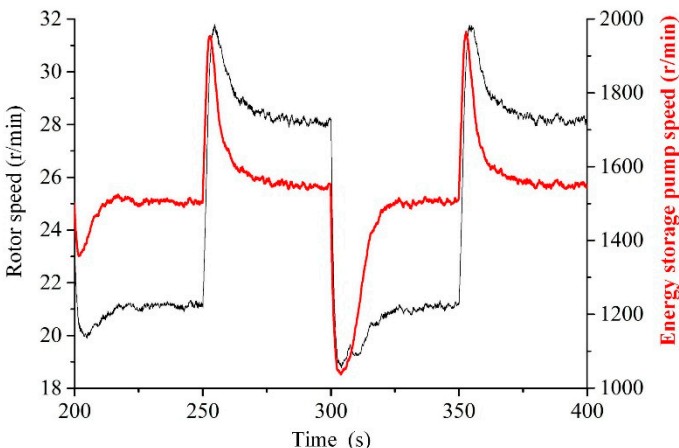

**Figure 15.** The speed curve of wind turbine rotor and energy storage pump.

5.2.2. Turbulent Wind Speed Condition

To implement more rigorously the maximum power tracking simulation experiment of the hydraulic energy storage wind turbine test rig, it is necessary to analyze the constitution of wind and use the actual and turbulent wind speed as the input signal to the simulation model. The turbulent wind speed curve is shown in Figure 16. During the entire wind speed simulation time of 100 s, the gradual wind and the gust play a major role in the wind speed change. In the simulated wind speed curve, the basic wind speed is 7 m/s, the maximum wind speed is 8 m/s, and the minimum wind speed is 6 m/s. The durations of multiple gradual winds and gusts are different.

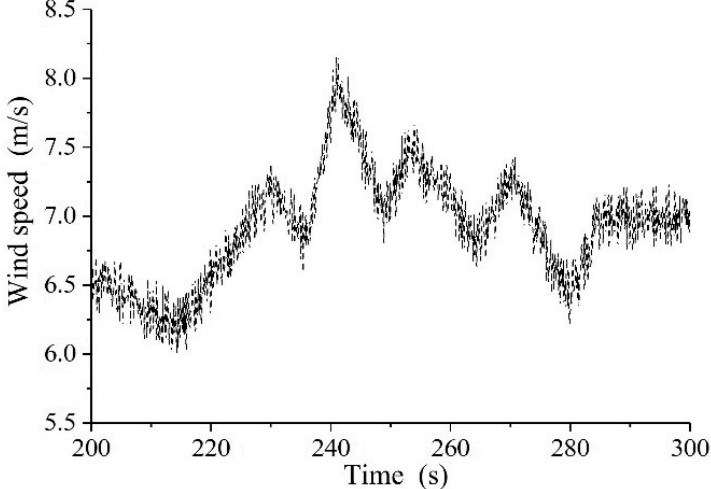

**Figure 16.** Turbulent wind speed.

It can be seen from Figure 17 that the actual speed of the wind turbine rotor under the maximum power tracking closed-loop control is exactly consistent with the optimum speed. Immediately following the optimal speed, the actual speed is slightly delayed. The wind turbine rotor rotates at the maximum and minimum speeds corresponding to the highest and lowest wind speeds, and the values are 27.3 r/min and 21.5 r/min, respectively.

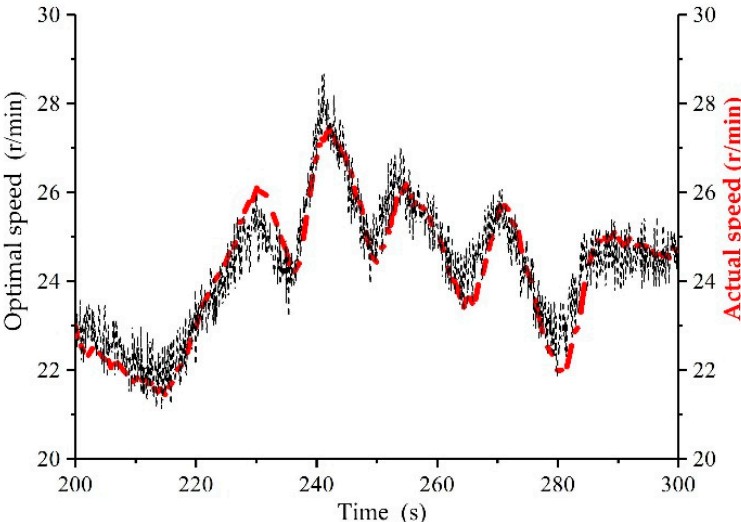

**Figure 17.** Comparison curve between wind turbine rotor actual speed and the optimal speed.

The displacement coefficients of the main motor and the energy storage pump are shown in Figure 18. Both displacement changes in real time and adjust in the same direction, so that the actual rotational speed of the wind turbine rotor can track the optimal input speed. Since the energy storage pump is the working load of the main motor, and the displacement adjustment of the energy storage pump brings about the load change of the main motor, which is an external disturbance to the energy storage pump speed control. In a word, the variation of wind speed is the fundamental reason that causes the speed fluctuation of the energy storage pump. As shown in Figure 19, the speed of the energy storage pump varies from 1400 r/min to 1600 r/min, which meets the speed requirements for the normal operation.

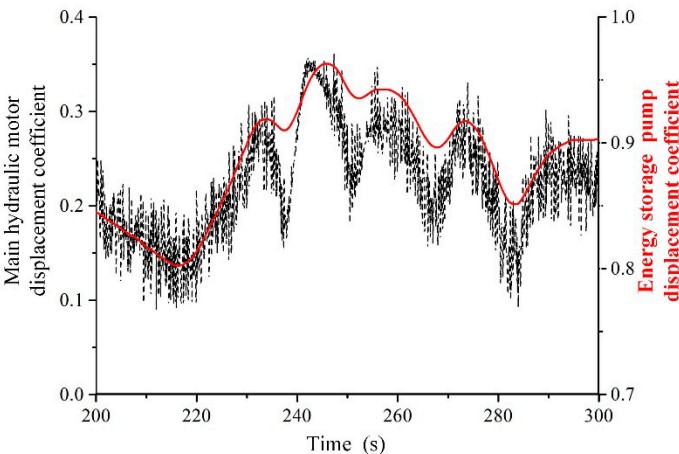

**Figure 18.** Displacement coefficient of energy storage pump and main hydraulic motor.

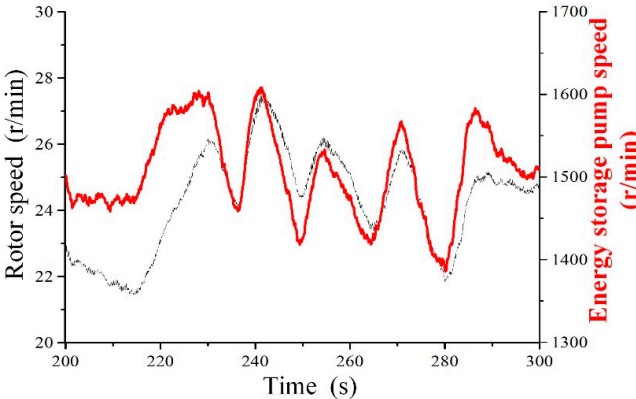

**Figure 19.** The speed curve of wind turbine rotor and energy storage pump.

It can be seen from Figure 20 that the magnitude of wind speed has an important influence on wind power. The higher the wind speed is, the greater the wind power. The wind power increased from 220 kW to 470 kW when the wind speed changed from 6 m/s to 8 m/s. The variation of the captured power by the wind turbine rotor is basically consistent with the change of the wind power. The maximum wind energy captured by wind turbine is about 200 kW and the minimum is approximately 90 kW. The power coefficient of the whole unit is shown in Figure 21. As shown in Figure 21, the power coefficient of the 600 kW hydraulic energy storage wind turbine fluctuates slightly around 0.44. The main reason for the fluctuation is the slight deviation between the actual wind turbine rotor speed and the input optimum speed.

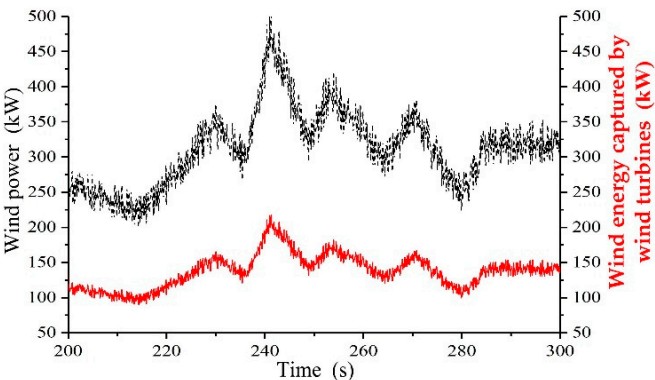

**Figure 20.** Comparison of wind energy and wind energy captured by wind turbine rotor.

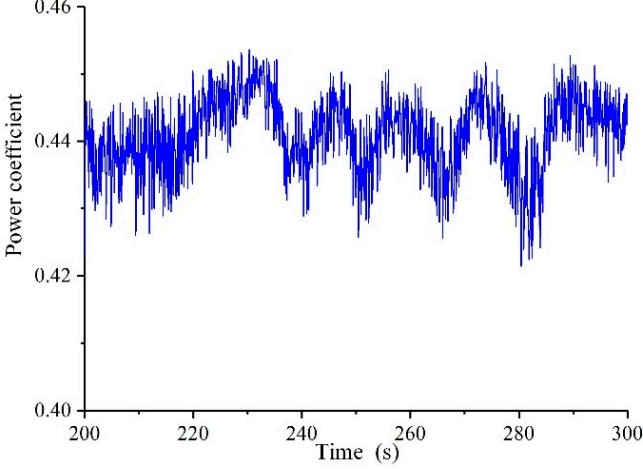

**Figure 21.** The curve of power coefficient.

## 6. Conclusions

The mechanical gearbox is changed to hydraulic drive for the Micon 600 wind turbine and the MPPT control method based on TSR is adopted to complete the maximum power extracting from wind energy. By comparing the aerodynamic torque of the wind turbine rotor with the mechanical counter torque produced by the energy storage pump, the speed control of the wind turbine rotor is realized. The control system that consists of three closed-loop controls is designed to obtain the maximum wind power absorption, wind energy storage, and constant frequency electrical energy.

The mathematical expressions of all components are derived, and the entire system model is constructed. Under both two different wind speed conditions, the simulation results show that the wind turbine rotor speed can track the optimal speed, but there is a small lag in time. The rotation speed of the energy storage pump can work well while the generator is running at the synchronous speed. The adjustment of the displacement coefficient of the energy storage hydraulic pump and the main motor plays an important role in controlling the rotational speed of the wind turbine rotor. In addition, the power coefficient of the wind turbine experimental prototype fluctuates around the optimal value, 0.44.

The proposed hydraulic wind turbine idea offers distinct advantages that one accumulator group can be shared by multiple wind turbines. Next, we will research on the superiority of these hydraulic wind turbine groups sharing the same storage system.

**Author Contributions:** L.W. conceived the research; Z.L. and P.Z. achieved the simulation model and control principle design; Z.L. and P.Z. completed the model verification and wrote the paper together; D.Y. and Y.T. performed the arrangement and analysis of the simulation data; L.W. and D.Y. critically revised the paper and provided many valuable and constructive suggestions.

**Funding:** This research was funded by the National Natural Science Fund Project of China (51765033), the Gansu Provincial Science and Technology Major Project of China (17ZD2GA010), the Gansu Provincial Natural Science Foundation of China (17JR5RA127), and the Gansu Provincial Natural Science Foundation of China (18JR3RA155).

**Conflicts of Interest:** The authors declared no potential conflicts of interest with respect to the research, authorship, and publication of this article.

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
