# Peer review of "Modeling and Analysis of Maximum Power Tracking of a 600 kW Hydraulic Energy Storage Wind Turbine Test Rig"

_processes, doi:10.3390/pr7100706_

Round 1
Reviewer 1 Report
It is necessary to revise the manuscript for the publication for the journal, Processes by considering the following comments.
1) In Figure 2, the units of m/s and kW for x- and y-axes are not necessary.
2) “Wind wheel” is not a common word in wind industry. It is recommended to revise the word of “wind wheel” as another word such as “wind turbine rotor” and “rotor.”
3) In Figure 21, the authors show the wind energy utilization coefficient. But this is also not commonly accepted word. It might be power coefficient or system efficiency. The authors need to revise it to the commonly accepted word in wind industry.
4) So many subscripts are used and a little bit complicated. The authors need to clarify the symbols using a section of Nomenclature.
5) Equations 3 and 4 are not suitable for all wind turbines and those are just ones of empirical relationship, therefore it is necessary to add an appropriate reference when the equations are described.
6) For applying the TSR-based MPPT algorithm, it is necessary to measure the wind speed, however it is very difficult to measure the incident wind speed. Therefore it is recommended to describe this TSR-based MPPT algorithm can be applied using other MPPT algorithms and the discussions are almost the same with this case.
7)In Figure 14, the authors compared the optimal rotor speed and the actual rotor speed, and they described that the actual rotor speed follows the optimal rotor speed with a certain level of overshoot. But they just compared the 20 seconds results, and it seems not to converge to the optimal rotor speed. Therefore it is necessary to show another results which can show that the actual rotor speed converges to the optimal rotor speed.
Reviewer 2 Report
Thank you for the good work. I have a few comments. 1- Generally, English and grammar need to be revised. many typos and syntax errors have been found in the paper. for example'' Section 2 details the system configuration and 89 function modules of the hydraulic energy storage wind turbine''. and '' Finally, the conclusion is drawn in Section 6.'' 2- The quality of the figures need to be improved greatly. 3- Please provide the tuning parameters of the PID in a table. Did the author use a PI or PID? 4- Please include a figure where the speed error is illustrated. 5- Please apply a stepping wind for more than 20 secs. preferably 50 secs to address the controller performance in transient and steady-state operation. 6- Is the figure 16 necessary. please consider removing it. 7- What kind of filters did the authors used in figure 19? 8- The conclusion does not reflect the actual contents of the paper, Please modify. 9- some of the references need to be updated. please consider works which are published at least the 18 months ago
